# The Impact of the COVID-19 Pandemic on Quality Education of the Medical Young Generation

**DOI:** 10.3390/ijerph20053953

**Published:** 2023-02-23

**Authors:** Daniela Roxana Matasariu, Ludmila Lozneanu, Iuliana Elena Bujor, Alexandra Elena Cristofor, Cristina Elena Mandici, Marcel Alexandru Găină, Cristinel Ștefănescu, Vasile Lucian Boiculese, Ioana Popescu, Laura Stătescu, Andreea Rusu, Simona Eliza Giusca, Alexandra Ursache

**Affiliations:** 1Department of Obstetrics and Gynecology, “Grigore T. Popa” University of Medicine and Pharmacy, 700115 Iaşi, Romania; 2Department of Obstetrics and Gynecology, Cuza Vodă Hospital, 700038 Iaşi, Romania; 3Department of Morpho-Functional Sciences I—Histology-Pathology, “Grigore T. Popa” University of Medicine and Pharmacy, 700115 Iasi, Romania; 4Psychiatry, Department of Medicine III, “Grigore T. Popa” University of Medicine and Pharmacy, 700115 Iasi, Romania; 5Biostatistics, Department of Preventive Medicine and Interdisciplinarity, “Grigore T. Popa” University of Medicine and Pharmacy, 700115 Iasi, Romania; 6Department of Dermatology, “Grigore T. Popa” University of Medicine and Pharmacy, 700115 Iasi, Romania

**Keywords:** COVID-19, medical, dental, education, student, teaching, stress, healthcare technology

## Abstract

(1) Generating the need to impose social distancing to reduce the spread of the virus, the COVID-19 pandemic altered the ways in which the teaching process normally happens. The aim of our study was to determine the impact of online teaching on medical students during this period. (2) Our study included 2059 medical, dental and pharmacy students from the University of Medicine and Pharmacy “Grigore T. Popa”, Iasi, Romania. We used a modified metacognition questionnaire after translation into Romanian and validation. Our questionnaire included 38 items, and it was divided into four parts. Academic results and preferences regarding the on-site or online courses, information regarding practical training, self-awareness in terms of one’s feelings such as anger, boredom and anxiety and also substance use linked to online teaching, and contextualization of the relationship with colleagues, teachers, friends and family were among the most important points evaluated. A comparison was made between preclinical and clinical students. A five-item Linkert-like scale was used for rating the answers in the last three parts that evaluated the impact of the SARS-CoV-2 pandemic on the educational process. (3) Preclinical medical students, compared to preclinical dental students, obtained statistically significant improvements in their evaluation results, with fewer failed exams (*p* < 0.001) and with similar results being obtained by comparing dental with pharmacy students. All students obtained statistically significant improvements in their academic results during the online evaluation. A statistically significant increase in anxiety and depression with a *p*-value of <0.001 was registered among our students. (4) The majority found it difficult to cope with this intense period. Both teachers and students found it difficult to adjust on such short notice to the challenges posed by the new concept of online teaching and learning.

## 1. Background

The COVID-19 pandemic utterly affected every aspect of our existence. The disease is caused by a Coronaviridae family virus (SARS-CoV-2), and was first reported in the city of Wuhan in Hubei Province, China at the end of December 2019 and beginning of January 2020. It manifests as a severe acute respiratory tract infection, and social distancing was adopted as a method to limit the spread of the virus [1].

In spite of the safety measures taken worldwide that were meant to protect the public’s health, people tended to become disconnected from their normal lives due to losing access to schools, workplaces, recreational places, and—not in a few cases—connections with friends and family [2]. Everyone experienced disruptions of their normal, usual life, with no access to public places. Losing connections with close ones had a major psychological and social impact [3]. Romania was no exception; it marked the start of the epidemic on 26 February 2020, with a state of emergency soon established in March [4]. The online environment was revised by the Ministry of Education as a solution to carry on with the teaching process in this context, and it became mandatory to continue school educational services in that setting [5].

In regards to the academic environment, the measures used for prophylaxis were mainly based on ending in-person classes, thus moving the didactic process online and trying to dissuade students from assembling on university campuses [6].

In the complex art of teaching, the most important effects of education worldwide can be boiled down on a basic level to obtaining good grades and academic success, by preserving an environment conducive to development for learners and obtaining positive outcomes for the educational system in its entirety. A major aspect to take into consideration would be to discover the factors that impact academic success and try to mold them in such a fashion that they reach a perfect balance permitting the optimal development of the student within that environment. Further, as we are talking about medical students— future doctors, a disruption in the learning environment could lead to an interesting palette of effects [7]. The COVID-19 pandemic represented the perfect setting for this disruption to happen.

In 2017, Chin et al. reported that a student’s positive emotions have a major impact on their academic performance and that alteration of the learning environment can lead to impaired outcomes [8].

Additionally, works in the medical literature regarding mental health found that medical students are exposed to stressful contexts in their learning environment, leading to high rates of anxiety and depression due to uncertainties and disruptions [6,9]. On the other hand, experiencing emotions such as hope, pride, enjoyment [10] and belongingness can lead to better final exam scores [11].

Generating the need to impose social distancing to reduce the spread of the virus, the COVID-19 pandemic altered the ways in which the teaching process normally happens. In a very short period of time, the professors found themselves in the position of elaborating and adopting a completely new online teaching system to ensure the continuity of learning. In the vast majority of domains, technology-based learning was implemented as the new “gold standard” in terms of teaching. Medical specialties were no exception but definitely faced a series of particularities in terms of clinical skills [12]. It was a very stressful situation for both teachers and students inexperienced with the novel online learning approach [13]. The sudden need to have a great variety of digital skills, competencies and resources could have also impacted the emotional, psychological and social well-being of students [14].

The University of Medicine and Pharmacy “Grigore T. Popa” from Iasi made no exception and offered the alternative of online teaching platform using Microsoft Teams. The new method was swiftly adopted, and both students and teachers had to adjust to it without extensive prior training [6].

Medical students felt the effects of these swift changes in the learning environment with both positive and negative implications. Uneven access to education arose as a serious problem, because the new learning environment implied previously acquired technical skills and access to a computer that could sustain a stable internet connection [6,15].

Thus, uneven access to the digital setting could be considered an underlying mechanism that led to lower academic results for low-income students that could not afford the necessary equipment for sustained online learning [16,17,18].

In terms of the impact of online teaching on medical students, the literature showed us that a qualitative design of the online content could help students to feel more motivated, with less dissatisfaction and a better capacity for learning. One problem that we need to take into consideration, and that is emerging from the change from a “face-to-face” to digital academic environment, is that while the so called “know-how” that translates into cognitive skills may be acquired in an online setting, students need to gain experience in clinical and behavioral skills through on-site settings that facilitate interaction with patients and in-person learning [15].

Students have also faced serious challenges because the online teaching and learning system demands more self-learning and self-discipline than traditional in-person learning [14].

Because few studies have attempted to evaluate the effects of the COVID-19 pandemic on medical students and their inner and outer environmental equilibrium, we decided to research these aspects. Only a limited number of studies have attempted to assess the effects of the COVID-19 pandemic on medical students. These studies concluded that the lockdown period disrupted medical students’ education during their clinical training years, but obtained conflicting results regarding psychological impairment [19,20,21,22,23,24,25]. The aim of our study was to determine the impact of online teaching on medical students during this period, by investigating educational and psychological aspects and specifically, the impact of online teaching on medical students’ physical and mental health. The study was carried out in the Grigore T. Popa University of Medicine and Pharmacy, Iasi, Romania, and included Romanian students from all three faculties: medicine, dentistry and pharmacy. No study on this topic had been developed in our university before.

## 2. Participants and Methods

The present paper was an observational study based on a questionnaire addressed to students.

Our study included 2059 medical (1547), dental (440) and pharmacy (72) students from the University of Medicine and Pharmacy “Grigore T. Popa”, Iasi. Providing responses to the questionnaire was assumed to indicate informed consent by each student to participate in the study. Student anonymity was maintained throughout the study, and students were assured their answers would be kept confidential.

The questionnaire was created and distributed using a digital application (Google Forms) and printed forms were provided to all medical, dental and pharmacy students from the University of Medicine and Pharmacy “Grigore T. Popa” Iasi, across all academic years. Given that first-year students were not studying at our University at the time due to pandemic-related restrictions, they were excluded from the study. Exclusion criteria were also included incomplete questionnaires or questionnaires submitted after the deadline. Erasmus program students were also excluded.

To be enrolled in the study, it was mandatory that students be in their second to sixth year of study in one of the three main domains mentioned above, at the University “Gr. T. Popa”, Iasi.

In addition, the questionnaire had to be submitted before the deadline. For submissions to be valid, all 38 items on the questionnaire had to be answered.

All of the participants were informed regarding the scope of the study. Consent to enroll in the study was assumed by simply completing the questionnaire. When the questionnaire was distributed, the University had been out of lockdown due to the SARS-CoV-2 pandemic for 3 months. The online forms were randomly distributed using social platforms and specific groups. The printed forms were given to students during classes and in stages. The questionnaire was approved for evaluation by the University Ethical Committee (No. 170 from 22.03.2022). We used a modified metacognition questionnaire after translation into Romanian and validation. The items were translated into the Romanian language by three translators, and we selected the best version for each item.

Our questionnaire included 38 items and evaluated the impact of the SARS-CoV-2 pandemic on the educational process. The selection of the items was made taking into account certain principles. First of all, the language used was accepted as appropriate to the socio-cultural environment of the subjects to whom the questionnaire was addressed. The items did not contain ambiguous wording, rarely used terms, multiple questions or answers at the same time. Items with negative or offensive wording were avoided as much as possible. The wording of the items was as short as possible but with discriminative power, i.e., it was possible to distinguish subjects that were positioned differently on the “range” of the scale.

The questionnaire contained four parts:-The first part collected data regarding the academic status of the students (general grade point average, number of failed exams) and also the general opinions of the students with reference to their preferences for online or on-site teaching in general (four items).-The second and third parts of the questionnaire included 12 and 13 items, respectively, and collected data regarding the online conduct of seminars and courses. With the help of these questions, we assessed metacognitive components such as knowledge, monitoring, planning, evaluation and control.-The last part focused on increased or newly developed dependencies (consumption of cigarettes, alcohol, and drugs) and the relationships of students with their friends, colleagues, and family members in the context of social distancing (nine items).-The questions for parts two to four were formulated in a self-rated way; the answers were quantified using a 5-point Likert-type scale, with 1 meaning complete disagreement with the statement, and 5 indicating complete agreement.

No rewards were offered to students, and the possibility of withdrawing from the research with no consequences, at any given time, was explained to them.

In total, 742 students accessed and completed the online survey anonymously, and 1317 completed it using the printed format. However, 941 returned questionnaires were not included in the study because they were submitted by students who were either not eligible, did not complete all questions, or did not submit their completed form by the deadline. Completion of the survey took approximately 20 min.

### Statistical Analysis

The findings were analyzed using the SPSS version 23 application (IBM SPSS, Chicago, IL, USA) for Microsoft Windows. Since the data were of categorical or ordinal types, we analyzed their frequency with the Chi-squared test. A *p*-value < 0.05 was considered to be significant. In tables, the symbol “!” flagged tests that were not consistent. These were defined as having more than 20% of expected frequencies less than 5.

## 3. Results

The survey form was sent to 3000 students who were registered across all years of study. A total of 2059 participants answered the questionnaire before the deadline.

### 3.1. Part I

Preclinical medical students obtained statistically significant improvements in their evaluation results compared to preclinical dental students, with fewer failed exams (429/195 and 85/135, *p* < 0.001), similarly to the results obtained when we compared dental students with pharmacy students (85/135 and 46/26, *p* < 0.001). The differences between preclinical medical students and pharmacy students were not statistically significant. Most medical students preferred online courses except when comparing preclinical medical students and preclinical dental students, where we noticed no statistically significant value (*p* = 0.423). However, in the evaluation of their preferences regarding practical training, the majority indicated that they found clinical practice with patient interaction more attractive.

### 3.2. Part II

In the second part of our questionnaire, we sought to evaluate students’ opinions regarding online practical stages that took place during the pandemic. As stated before, in the first part, most students found on-site clinical practice with patient interaction more resourceful for acquiring practical skills. They disagreed with the statement that online practical stages helped them save time, improve their attendance, understand concepts and obtain better academic results. The differences were statistically significant when comparing preclinical with clinical medical students, preclinical medical students with dental students, clinical medical students with clinical dental students, and preclinical medical students with pharmacy students. The results were not statistically significant when we compared preclinical dental students with pharmacy students regarding their preferences for online practical stages and the negative impact of online clinical training on the acquisition of practical skills. However, compared with pharmacy students, medical and dental students—with the exception of their preclinical counterparts—showed statistically significant agreement regarding the importance of learning during stages, declaring that online clinical practice evaluations did not add extra stress. All students obtained statistically significant improvements in their academic results during online practical stage evaluations, despite declaring that they did not obtain more information through this method of teaching (Table 1).

### 3.3. Part III

When students were evaluated about their opinions of online courses, their preferences changed; most students agreed that online courses were a better way of teaching, being more time-saving. They declared success in accumulating knowledge and communicating with their teachers. When evaluated about the ability of teachers to maintain their attention and deliver information, most students agreed that their teachers were successful in these regards. They did not perceive online courses as boring or time-wasting. They were in statistically significant agreement that it is important to learn new concepts during courses and did not perceive the online evaluations as inflicting extra stress. Due to the fact that in dental medicine, the distribution of actual clinical practice is more balanced across one’s student years, with general clinical subjects studied in the last three years, differences in academic performance were not statistically significant, but when we evaluated the answers from medical students, we discovered that clinical medical students did not obtain better academic results during their evaluations due to a lack of actual clinical practice that would have helped them better learn and remember new concepts (Table 2).

### 3.4. Part IV

Our medical students did not declare developing or increasing addictive behaviors (alcohol, smoking or using drugs) during the pandemic, but visible and statistically significant increases in anxiety and depression were reported, with a *p* value of <0.001. The only relationships that seemed to be affected during social distancing were those with their colleagues, especially among the preclinical students. When evaluated about returning to on-site teaching and learning, most students declared that they had not encountered any difficulties (Table 3).

## 4. Discussion

This study evaluated the impact of physical course suspension on clinical and preclinical learning, comparing academic outcomes between the three faculties of medicine, dentistry and pharmacy as well as on students’ mental health by assessing potential psychological distress associated with specific addictions such as drug use between the on-site and online learning periods during the COVID-19 pandemic. This study is the first to investigate this issue in students of “Grigore T. Popa” University of Iasi, Romania. Our assumptions were partially confirmed, showing that undergraduate students in all three faculties performed at least equally well in both clinical and preclinical settings. However, their mental health was only impacted due to the restrictions on physical socialization during that lockdown period; meeting with peers was not possible.

Online teaching and study offered the possibility of continuing the learning process in the pandemic context, where social distancing was mandatory to limit the spread of the disease. While remote learning assured access to information, training in practical skills remained deficient [6].

Planas et al. underlined in his study that less favorable academic results might stand as an expression of low-income students’ uneven access to technology and higher stress and anxiety due to learning environment alterations, with disruptions and major transformations [18]. The literature abounds with evidence that the COVID-19 pandemic, along with the technological changes and online shifts that accompanied it, had a massive impact in widening the socio-economic educational gap [26,27,28]. Even though Romania is a middle-income country, with many students coming from less favorable environments and saddled with financial difficulties, our study revealed that online courses were preferred and contributed to increases in academic performance. These aspects might suggest that either online courses were better explained and presented by teachers, with fast access to alternative interactive means of teaching, or that the evaluations by the students were less stringent than before pandemic. Using comparative studies between online and face-to-face versions of the same course for a period of 8 years, the U.S. Department of Education concluded that online learning produces learning outcomes similar to or better than those of face-to-face learning [7,29]. Even if many students favored online courses in the context of practical training and stages, they agreed that the on-site method, which assures interaction with patients, offered them better grounding. The possibility of developing medical critical thinking and practical skills was altered during pandemic online teaching. A survey that addressed students’ opinions regarding the subject of online vs. on-site learning found that 65.5% were happier with the online alternative, although many of them felt unsatisfied with online teaching in terms of practical skills development [30].

Online courses were also preferred due to the simple fact that they helped students save a significant amount of time required to physically attend classes; many survey respondents admitted that they attended more classes online than on-site. When we compared the responses of preclinical medical students (students in the second and third year of the medicine program) to those of clinical medical students (students in the last three years of the medicine program), it became obvious that the latter group valued the application of the knowledge they had acquired through lectures. AlQhtani et al. concluded that the best and also practical solution for maintaining a high-quality level of teaching and learning is the hybrid one, also called “blended teaching”, that combines the advantages of the two given methods [30].

For dental and pharmacy students, these aspects were less apparent, due to the fact that their clinical skills and training are less patient interaction-dependent. Iosif et al. concluded in their study that for dental students, the lack of practice had a serious impact on their manual dexterity and fine motor skills. Additionally, they observed that the students most impacted by this problem were those in the clinical study years 4, 5 and 6 [31]. Furthermore, a study that surveyed dental students from 34 countries found suggestive evidence that students who were in the so-called “early clinical years” with less contact with patients had lower odds of being satisfied with online courses than senior students [15].

In a 2010 study that evaluated the connection between medical students’ emotions and their mental health, Artino et al. underlined that many students experience stressful situations in their learning environment that cause depression and anxiety [9]. In terms of quarantine and its effects on their psycho-emotional and social behavior, both male and female students declared that they felt estranged from family and friends. Additionally, their intellectual performance declined, and one quarter of the students felt depressed [32]. The COVID-19 pandemic created even more new stressful situations in the learning environment for medical students, adding the need to almost instantly acquire digital skills and resources [14,33]. A Spanish study that evaluated the impact of the COVID-19 pandemic showed that levels of depression, stress and anxiety among medical students were over the limit [34]. In fact, most studies identified higher levels of psychological stress among students during university closures [14,25,33,35,36]. The present study did not find a difference in psychological distress during periods of physical versus online learning. These conflicting results may be due to the time period in which the study was conducted. Existing research that has shown psychological distress among students during isolation or school closure examined this issue in the first wave of the COVID-19 pandemic. Certainly, the level of stress at the time was due not only to changes in education but also to the uncertain situation we were all facing. Thus, as mentioned earlier, this study was conducted at a time when the students had already been through this situation and thus had experienced the changes induced by the other COVID-19 outbreaks. Therefore, in this study, no deterioration in mental health was found, only the need to meet and socialize with peers.

Cunha et al. emphasized in a 2006 study that another stress-generating factor that may heavily impact a medical student’s academic performance is the fact that online learning requires greater self-discipline and effort to accumulate information by the students themselves, even if it offers the advantage of flexibility and quicker access to study materials. Both preclinical medical and dental students did not expect to obtain better results after online practical stages. However, it become obvious that students in their final years of study succeeded in gaining a better understanding of the concepts through online teaching, with better academic results due to the fact that they had developed better self-discipline, understanding, and connected concepts and learning skills during the time spent in their studies online. Further, students in the final years of study seemed more stressed during online examinations, had greater expectations, became uncomfortable and anxious when they lacked practical skills for their exams, were concerned with learning and accumulating information during classes, and more aware of the need to be academically prepared. Due to anxiety, the majority of the interviewed medical, dental and pharmacy students found it less stressful to ask questions and be examined online [37,38,39].

Another challenge for teachers during online learning is that of maintaining the focus and attention of their students by recreating the dynamics and interactions of face-to-face teaching without the aid of non-verbal communication [38,40]. In online teaching, it is very hard to build relationships with your students and between your students in order to foster academic success. The need for both teachers and students to adapt to the new way of teaching and learning involved significant stress for both [38]. We did not detect these aspects in our study; students did not declare any difficulties in concentration or increased anxiety and anger during online teaching, nor did they perceive online instruction to be a waste of time [40].

Many studies support the finding that prolonged exposure to display screens can seriously and negatively impact brain development, with cognitive, motor, learning, memory and emotional consequences. In part four of our questionnaire, we tried to evaluate students’ emotions, the development or progression of their addictions (smoking, drinking, and drug abuse) and the alterations in their relationships (with colleagues, teachers and family). We observed in our study that students did not manifest increased addictive behaviors such as smoking, drinking and drug abuse during the COVID-19 pandemic. Additionally, many of the medical students, and especially those in the preclinical stages of their medical and dental programs, declared an important impact of isolation with online teaching on their relationships with colleagues, but with no impact on relationships with family or friends. Social isolation exacerbates anxiety and depression, leading to substance, cigarette or alcohol abuse [41]. Solid and well-established relationships with family and friends might explain the lack of addictive behavior in our cohort of students.

## 5. Conclusions

The lockdown period manifested itself in divergent ways for students from medical departments. Some discovered a new way to develop their knowledge and to grow, due to online teaching that afforded the advantages of saving time and recording courses, and thus succeeded in achieving self-discipline during this difficult period. However, the majority found it difficult to cope with this intense period characterized by uncertainty, psychological distress, great anxiety, and fear. Both teachers and students found it difficult to adjust on such short notice to the challenges posed by the new concept of online teaching and learning. Nevertheless, they managed to disentangle the challenges together, through teacher–student collaboration and cooperation.

Our results suggest that the increase in feelings of anxiety and depression among students during the COVID-19 pandemic was associated not with the shift from physical to online learning, but rather with an inability to relate to their peers. Academic results were not adversely influenced, either; on the contrary, there was an increased level of progress among pre-clinical students. Students—and especially dental students—in the final years of study were the most affected due to the impossibility of physically attending training courses in hospitals, although to a lesser extent when the hybrid form of education was introduced. This explained their desire to return to the physical, in-person forms of instruction and internship. Despite these shortcomings, all students adapted smoothly to the new rules imposed by the transition to online learning. This shows that medical students are highly adaptable and can cope with changes of any kind.

The limitations of this study are that the survey was also conducted online, and physical recruitment was not possible in all cases. This reduced our ability to ensure the quality of responses from all participants. For example, some students completed part of the survey or omitted only a few items from the questionnaire. This led us to exclude some data for further analysis. Another limitation is that we initiated this study at a time when schools had been closed for 3 months, but this had been experienced previously, which may have caused erroneous responses among participants regarding their mental health due to their recall of that period. It would be desirable for pre-university and other educational institutions to regularly assess the mental health of students in the long term, so that we could more accurately assess the long-term impact of COVID-19 on mental health. Another limitation of our study is its small geographical scope, as the respondent students represented only one university.

## Figures and Tables

**Table 1 ijerph-20-03953-t001:** The questionnaire—part II.

	Medical Preclinic (years II, III)	Medical Clinic (years IV, V, VI)	*p* Value	Dental Preclinic (II, III)	Dental Clinic (IV, V, VI)	*p* Value	*p* Value Preclinic Medical/Dental	*p* Value Clinic Medical/Dental	Pharmacy	*p* Value Preclinic Medical/Pharmacy	*p* Value Preclinic Dental/Pharmacy
1. I prefer on-site clinical stages because they imply patient interaction that allows me to succeed in learning new things and acquire practical skills.											
1	26 (4.2%)	13 (1.4%)	<0.001	0 (0%)	0 (0%)	<0.001	<0.001	<0.001	0 (0.0%)	<0.001	!
2	0 (0%)	0 (0%)		0 (0.0%)	10 (4.5%)				4 (5.6%)		
3	39 (6.3%)	78 (8.5%)		0 (0.0%)	8 (3.6%)				4 (5.6%)		
4	65 (10.4%)	169 (18.3%)		40 (18.2%)	31 (14.1%)				4 (5.6%)		
5	494 (79.2%)	663 (71.8%)		180 (81.8%)	171 (77.7%)				60 (83.3%)		
2. I prefer online clinical stages because they are less time-consuming in terms of attendance and transportation to the university.											
1	338 (54.2%)	377 (40.3%)	<0.001	160 (72.7%)	112 (50.9%)	<0.001	<0.001	<0.001	57 (79.2%)	<0.001	<0.001
2	104 (16.7%)	247 (26.4%)		60 (27.3%)	45 (20.5%)				7 (9.7%)		
3	78 (12.5%)	169 (18.1%)		0 (0.0%)	25 (11.4%)				8 (11.1%)		
4	39 (6.3%)	91 (9.7%)		0 (0.0%)	8 (3.6%)				0 (0.0%)		
5	65 (10.4%)	52 (5.6%)		0 (0.0%)	30 (13.6%)				0 (0.0%)		
3. I attended more online clinical stages than I could have if attending on-site ones.											
1	351 (56.3%)	338 (36.6%)	<0.001	114 (51.8%)	102 (46.4%)	0.001	<0.001	0.007	50 (69.4%)	<0.001	0.003
2	52 (8.3%)	143 (15.5%)		36 (16.4%)	41 (18.6%)				10 (13.9%)		
3	156 (25.0%)	117 (12.7%)		34 (15.5%)	24 (10.9%)				4 (5.6%)		
4	39 (6.3%)	104 (11.3%)		22 (10.0%)	12 (5.5%)				0 (0.0%)		
5	26 (4.2%)	221 (23.9%)		14 (6.4%)	41 (18.6%)				8 (11.1%)		
4. I expect to obtain better results during online clinical stages.											
1	351 (56.3%)	390 (42.3%)	<0.001	133 (60.5%)	92 (41.8%)	<0.001	<0.001	<0.001	46 (63.9%)	<0.001	<0.001
2	78 (12.5%)	169 (18.3%)		50 (22.7%)	30 (13.6%)				4 (5.6%)		
3	130 (20.8%)	247 (26.8%)		25 (11.4%)	46 (20.9%)				4 (5.6%)		
4	39 (6.3%)	13 (1.4%)		0 (0.0%)	20 (9.1%)				8 (11.1%)		
5	26 (4.2%)	104 (11.3%)		12 (5.5%)	32 (14.5%)				10 (13.9%)		
5. I find it hard to understand the main ideas from an online clinical stage.											
1	143 (22.9%)	273 (29.6%)	<0.001	34 (15.5%)	67 (30.5%)	<0.001	<0.001	0.442	9 (12.5%)	<0.001	<0.001
2	52 (8.3%)	130 (14.1%)		32 (14.5%)	23 (10.5%)				7 (9.7%)		
3	117 (18.8%)	156 (16.9%)		60 (27.3%)	43 (19.5%)				2 (2.8%)		
4	130 (20.8%)	156 (16.9%)		0 (0.0%)	43 (19.5%)				29 (40.3%)		
5	182 (29.2%)	208 (22.5%)		94 (42.7%)	44 (20.0%)				25 (34.7%)		
6. I find it more comfortable to ask questions needed to understand new information during online clinical stages.											
1	286 (45.8%)	377 (40.8%)	<0.001	84 (38.2%)	90 (40.7%)	0.001	<0.001	0.001	37 (51.4%)	0.651	0.001
2	65 (10.4%)	169 (18.3%)		51 (23.2%)	20 (9.0%)				8 (11.1%)		
3	130 (20.8%)	156 (16.9%)		22 (10.0%)	34 (15.4%)				16 (22.2%)		
4	78 (12.5%)	91 (9.9%)		20 (9.1%)	33 (14.9%)				7 (9.7%)		
5	65 (10.4%)	130 (14.1%)		43 (19.5%)	44 (19.9%)				4 (5.6%)		
7. I consider that the lack of contact with patients during online clinical stages has a negative impact on my professional development (examination/direct visualization helps me to understand and remember concepts better).											
1	39 (6.3%)	39 (4.2%)	0.028	19 (8.6%)	20 (9.1%)	<0.001	<0.001	<0.001	4 (5.6%)	<0.001	!
2	26 (4.2%)	26 (2.8%)		0 (0%)	0 (0%)				3 (4.2%)		
3	39 (6.3%)	78 (8.5%)		0 (0.0%)	31 (14.1%)				16 (22.2%)		
4	91 (14.6%)	169 (18.3%)		17 (7.7%)	15 (6.8%)				9 (12.5%)		
5	429 (68.8%)	611 (66.2%)		184 (83.6%)	154 (70.0%)				40 (55.6%)		
8. I feel ashamed during exams when I discover that I lack certain clinical skills.											
1	78 (12.5%)	39 (4.2%)	0.001	0 (0.0%)	8 (3.6%)	0.039	<0.001	0.018	13 (18.1%)	0.011	!
2	13 (2.1%)	13 (1.4%)		0 (0%)	0 (0%)				5 (6.9%)		
3	65 (10.4%)	104 (11.3%)		41 (18.6%)	41 (18.6%)				1 (1.4%)		
4	117 (18.8%)	221 (23.9%)		51 (23.2%)	45 (20.5%)				13 (18.1%)		
5	351 (56.3%)	546 (59.2%)		128 (58.2%)	126 (57.3%)				40 (55.6%)		
9. I feel less stress during online exams (regarding concepts learned during clinical stages).											
1	169 (27.1%)	273 (29.6%)	<0.001	45 (20.5%)	45 (20.5%)	0.464	0.267	<0.001	21 (29.2%)	0.006	0.020
2	26 (4.2%)	130 (14.1%)		10 (4.5%)	11 (5.0%)				10 (13.9%)		
3	91 (14.6%)	169 (18.3%)		28 (12.7%)	21 (9.5%)				7 (9.7%)		
4	156 (25.0%)	234 (25.4%)		62 (28.2%)	78 (35.5%)				13 (18.1%)		
5	182 (29.2%)	117 (12.7%)		75 (34.1%)	65 (29.5%)				21 (29.2%)		
10. I find it very important to learn during the study year the concepts explained during clinical stages.											
1	0 (0.0%)	26 (2.8%)	<0.001	0 (0%)	0 (0%)	0.006	<0.001	<0.001	0 (0.0%)	0.122	0.799
2	26 (4.2%)	78 (8.5%)		0 (0.0%)	11 (5.0%)				0 (0.0%)		
3	65 (10.4%)	117 (12.7%)		36 (16.4%)	38 (17.3%)				12 (16.7%)		
4	104 (16.7%)	286 (31.0%)		51 (23.2%)	40 (18.2%)				14 (19.4%)		
5	429 (68.8%)	416 (45.1%)		133 (60.5%)	131 (59.5%)				46 (63.9%)		
11. I obtained better academic results during the final online clinical stage evaluation.											
1	130 (20.8%)	273 (29.6%)	<0.001	63 (28.6%)	29 (12.1%)	<0.001	<0.001	<0.001	20 (27.8%)	0.034	<0.001
2	52 (8.3%)	117 (12.7%)		51 (23.2%)	8 (3.3%)				8 (11.1%)		
3	221 (35.4%)	156 (16.9%)		74 (33.6%)	53 (22.1%)				12 (16.7%)		
4	91 (14.6%)	143 (15.5%)		9 (4.1%)	58 (24.2%)				12 (16.7%)		
5	130 (20.8%)	234 (25.4%)		23 (10.5%)	92 (38.3%)				20 (27.8%)		
12. I succeeded in acquiring more information during online clinical stages.											
1	123 (24.0%)	377 (40.8%)	<0.001	97 (44.1%)	84 (38.2%)	<0.001	<0.001	<0.001	39 (55.7%)	0.017	!
2	169 (32.9%)	221 (23.9%)		65 (29.5%)	9 (4.1%)				16 (22.9%)		
3	130 (25.3%)	169 (18.3%)		51 (23.2%)	80 (36.4%)				9 (12.9%)		
4	52 (10.1%)	78 (8.5%)		0 (0.0%)	31 (14.1%)				6 (8.6%)		
5	39 (7.6%)	78 (8.5%)		7 (3.2%)	16 (7.3%)				0 (0.0%)		

*p*-value < 0.05 was considered to be significant. !—tests not consistent.

**Table 2 ijerph-20-03953-t002:** Part III of questionnaire.

	Medical Preclinic (years II, III)	Medical Clinic (years IV, V, VI)	*p* Value	Dental Preclinic (II, III)	Dental Clinic (IV, V, VI)	*p* Value	*p* Value Preclinic Medical/Dental	*p* Value Clinic Medical/Dental	Pharmacy	*p* Value Preclinic Medical/Pharmacy	*p* Value Preclinic Dental/Pharmacy
1. I like attending courses.											
1	65 (10.4%)	156 (16.9%)	<0.001	9 (4.1%)	59 (26.8%)	<0.001	<0.001	<0.001	10 (13.9%)	0.015	!
2	65 (10.4%)	91 (9.9%)		0 (0.0%)	41 (18.6%)				14 (19.4%)		
3	143 (22.9%)	299 (32.4%)		92 (41.8%)	70 (31.8%)				7 (9.7%)		
4	234 (37.5%)	221 (23.9%)		62 (28.2%)	29 (13.2%)				23 (31.9%)		
5	117 (18.8%)	156 (16.9%)		57 (25.9%)	21 (9.5%)				18 (25.0%)		
2. I attended more online courses than I could have if attending on-site ones.											
1	104 (16.7%)	91 (9.9%)	<0.001	29 (13.2%)	0 (0.0%)	<0.001	<0.001	<0.001	28 (38.9%)	<0.001	!
2	39 (6.3%)	65 (7.0%)		9 (4.1%)	8 (3.6%)				0 (0.0%)		
3	91 (14.6%)	65 (7.0%)		8 (3.6%)	7 (3.2%)				3 (4.2%)		
4	143 (22.9%)	91 (9.9%)		52 (23.6%)	9 (4.1%)				9 (12.5%)		
5	247 (39.6%)	611 (66.2%)		122 (55.5%)	196 (89.1%)				32 (44.4%)		
3. I prefer online courses because they are less time-consuming in terms of attendance and transportation to the university.											
1	104 (16.7%)	39 (4.2%)	<0.001	32 (14.5%)	8 (3.6%)	<0.001	<0.001	<0.001	25 (34.7%)	<0.001	!
2	39 (6.3%)	26 (2.8%)		9 (4.1%)	8 (3.6%)				0 (0.0%)		
3	104 (16.7%)	78 (8.5%)		7 (3.2%)	14 (6.4%)				6 (8.3%)		
4	65 (10.4%)	78 (8.5%)		42 (19.1%)	1 (0.5%)				4 (5.6%)		
5	312 (50.0%)	702 (76.1%)		130 (59.1%)	189 (85.9%)				37 (51.4%)		
4. I find it hard to understand the main ideas from an online course.											
1	234 (37.5%)	572 (62.0%)	<0.001	42 (19.1%)	112 (50.9%)	<0.001	<0.001	<0.001	33 (45.8%)	0.133	<0.001
2	104 (16.7%)	143 (15.5%)		82 (37.3%)	58 (26.4%)				12 (16.7%)		
3	130 (20.8%)	143 (15.5%)		51 (23.2%)	30 (13.6%)				11 (15.3%)		
4	39 (6.3%)	39 (4.2%)		37 (16.8%)	20 (9.1%)				0 (0.0%)		
5	117 (18.8%)	26 (2.8%)		8 (3.6%)	0 (0.0%)				16 (22.2%)		
5. I find it more comfortable to ask questions needed to understand new information during online courses.											
1	169 (27.1%)	143 (15.5%)	<0.001	29 (13.2%)	35 (15.9%)	0.004	<0.001	0.137	20 (27.8%)	0.506	0.001
2	104 (16.7%)	195 (21.1%)		63 (28.6%)	32 (14.5%)				8 (11.1%)		
3	143 (22.9%)	195 (21.1%)		36 (16.4%)	42 (19.1%)				20 (27.8%)		
4	78 (12.5%)	130 (14.1%)		41 (18.6%)	37 (16.8%)				12 (16.7%)		
5	130 (20.8%)	260 (28.2%)		51 (23.2%)	74 (33.6%)				12 (16.7%)		
6. My anger increases when I think about the time wasted during online courses.											
1	286 (45.8%)	585 (63.4%)	<0.001	118 (53.6%)	120 (54.5%)	0.067	<0.001	<0.001	52 (72.2%)	<0.001	0.034
2	91 (14.6%)	182 (19.7%)		54 (24.5%)	47 (21.4%)				8 (11.1%)		
3	208 (33.3%)	52 (5.6%)		29 (13.2%)	44 (20.0%)				8 (11.1%)		
4	13 (2.1%)	26 (2.8%)		19 (8.6%)	9 (4.1%)				4 (5.6%)		
5	26 (4.2%)	78 (8.5%)		0 (0%)	0 (0%)				0 (0.0%)		
7. I wish I were not constrained to attend online courses because they irritate me.											
1	299 (47.9%)	624 (67.6%)	<0.001	151 (68.6%)	132 (62.9%)	<0.001	<0.001	<0.001	35 (48.6%)	<0.001	0.001
2	78 (12.5%)	117 (12.7%)		32 (14.5%)	21 (10.0%)				17 (23.6%)		
3	104 (16.7%)	39 (4.2%)		0 (0%)	0 (0.0%)				0 (0.0%)		
4	78 (12.5%)	104 (11.3%)		15 (6.8%)	57 (27.1%)				15 (20.8%)		
5	65 (10.4%)	39 (4.2%)		22 (10.0%)	0 (0.0%)				5 (6.9%)		
8. Online courses bore me.											
1	195 (31.3%)	403 (43.7%)	<0.001	75 (34.1%)	96 (43.6%)	0.049	<0.001	0.019	24 (33.3%)	0.011	<0.001
2	78 (12.5%)	182 (19.7%)		45 (20.5%)	40 (18.2%)				0 (0.0%)		
3	117 (18.8%)	104 (11.3%)		20 (9.1%)	29 (13.2%)				20 (27.8%)		
4	117 (18.8%)	143 (15.5%)		25 (11.4%)	20 (9.1%)				11 (15.3%)		
5	117 (18.8%)	91 (9.9%)		55 (25.0%)	35 (15.9%)				17 (23.6%)		
9. I often think of what else I can do with the time I waste attending online courses.											
1	234 (37.5%)	520 (56.3%)	<0.001	74 (33.6%)	113 (51.4%)	<0.001	<0.001	<0.001	30 (41.7%)	0.014	<0.001
2	78 (12.5%)	182 (19.7%)		24 (10.9%)	42 (19.1%)				12 (16.7%)		
3	104 (16.7%)	104 (11.3%)		24 (10.9%)	12 (5.5%)				5 (6.9%)		
4	91 (14.6%)	13 (1.4%)		84 (38.2%)	12 (5.5%)				4 (5.6%)		
5	117 (18.8%)	104 (11.3%)		14 (6.4%)	41 (18.6%)				21 (29.2%)		
10. I am impatient and wish that online courses would end faster.											
1	221 (35.4%)	481 (52.1%)	<0.001	57 (25.9%)	72 (34.3%)	<0.001	<0.001	<0.001	22 (26.2%)	0.14	<0.001
2	65 (10.4%)	169 (18.3%)		50 (22.7%)	33 (15.7%)				6 (7.1%)		
3	104 (16.7%)	39 (4.2%)		16 (7.3%)	70 (33.3%)				15 (17.9%)		
4	65 (10.4%)	91 (9.9%)		31 (14.1%)	13 (6.2%)				4 (4.8%)		
5	169 (27.1%)	143 (15.5%)		66 (30.0%)	22 (10.5%)				37 (44.0%)		
11. I find it very important to learn during the study year the concepts explained during courses.											
1	26 (4.2%)	91 (9.9%)	<0.001	13 (5.9%)	10 (4.5%)	<0.001	<0.001	<0.001	0 (0.0%)	<0.001	!
2	0 (0.0%)	117 (12.7%)		7 (3.2%)	0 (0.0%)				8 (11.1%)		
3	104 (16.7%)	169 (18.3%)		31 (14.1%)	150 (68.2%)				12 (16.7%)		
4	273 (43.8%)	195 (21.1%)		96 (43.6%)	33 (15.0%)				14 (19.4%)		
5	221 (35.4%)	351 (38.0%)		73 (33.2%)	27 (12.3%)				38 (52.8%)		
12. I feel less stress during online exams (regarding concepts learned during courses).											
1	208 (33.3%)	377 (40.8%)	<0.001	51 (23.2%)	34 (15.5%)	<0.001	0.002	<0.001	28 (38.9%)	0.196	0.053
2	52 (8.3%)	117 (12.7%)		31 (14.1%)	0 (0.0%)				8 (11.1%)		
3	182 (29.2%)	143 (15.5%)		55 (25.0%)	62 (28.2%)				12 (16.7%)		
4	104 (16.7%)	169 (18.3%)		41 (18.6%)	50 (22.7%)				16 (22.2%)		
5	78 (12.5%)	117 (12.7%)		42 (19.1%)	74 (33.6%)				8 (11.1%)		
13. I obtained better academic results during online exams.											
1	91 (14.6%)	273 (29.6%)	<0.001	65 (29.5%)	30 (13.6%)	<0.001	<0.001	<0.001	16 (22.2%)	0.002	<0.001
2	78 (12.5%)	130 (14.1%)		26 (11.8%)	9 (4.1%)				16 (22.2%)		
3	208 (33.3%)	195 (21.1%)		70 (31.8%)	81 (36.8%)				16 (22.2%)		
4	117 (18.8%)	156 (16.9%)		36 (16.4%)	18 (8.2%)				4 (5.6%)		
5	130 (20.8%)	169 (18.3%)		23 (10.5%)	82 (37.3%)				20 (27.8%)		

*p*-value < 0.05 was considered to be significant. !—tests not consistent.

**Table 3 ijerph-20-03953-t003:** Part IV of questionnaire.

	Medical Preclinic (years II, III)	Medical Clinic (years IV, V, VI)	*p* Value	Dental Preclinic (II, III)	Dental Clinic (IV, V, VI)	*p* Value	*p* Value Preclinic Medical/Dental	*p* Value Clinic Medical/Dental	Pharmacy	*p* Value Preclinic Medical/Pharmacy	*p* Value Preclinic Dental/Pharmacy
1. Did you increase your alcohol consumption during the pandemic?											
1	468 (75.0%)	650 (70.5%)	<0.001	151 (68.6%)	151 (68.6%)	<0.001	<0.001	<0.001	60 (83.3%)	0.005	!
2	91 (14.6%)	130 (14.1%)		20 (9.1%)	39 (17.7%)				0 (0.0%)		
3	26 (4.2%)	116 (12.6%)		37 (16.8%)	9 (4.1%)				4 (5.6%)		
4	13 (2.1%)	13 (1.4%)		0 (0.0%)	10 (4.5%)				4 (5.6%)		
5	26 (4.2%)	13 (1.4%)		12 (5.5%)	11 (5.0%)				4 (5.6%)		
2. Did you start smoking or smoke more during the pandemic?											
1	520 (83.3%)	689 (74.6%)	<0.001	187 (85.0%)	181 (82.3%)	<0.001	<0.001	0.013	56 (77.8%)	0.103	!
2	52 (8.3%)	117 (12.7%)		13 (5.9%)	23 (10.5%)				4 (5.6%)		
3	26 (4.2%)	78 (8.5%)		0 (0.0%)	7 (3.2%)				5 (6.9%)		
4	0 (0.0%)	13 (1.4%)		12 (5.5%)	0 (0.0%)				0 (0.0%)		
5	26 (4.2%)	26 (2.8%)		8 (3.6%)	9 (4.1%)				7 (9.7%)		
3. Did you start abusing substances (drugs) during the pandemic?											
1	585 (93.8%)	884 (95.8%)	<0.001	210 (95.5%)	220 (100.0%)	0.001	<0.001	0.008	68 (94.4%)	0.103	!
2	13 (2.1%)	26 (2.8%)		0 (0.0%)	0 (0.0%)				0 (0.0%)		
3	13 (2.1%)	0 (0.0%)		0 (0.0%)	0 (0.0%)				4 (5.6%)		
4	0 (0.0%)	0 (0.0%)		10 (4.5%)	0 (0.0%)				0 (0.0%)		
5	13 (2.1%)	13 (1.4%)		0 (0.0%)	0 (0.0%)				0 (0.0%)		
4. Did you more frequently display signs of depression (anxiety, crying)?											
1	117 (18.8%)	364 (34.6%)	<0.001	49 (22.3%)	87 (39.5%)	<0.001	<0.001	<0.001	12 (16.7%)	<0.001	<0.001
2	91 (14.6%)	182 (17.3%)		27 (12.3%)	28 (12.7%)				0 (0.0%)		
3	208 (33.3%)	130 (12.3%)		19 (8.6%)	7 (3.2%)				8 (11.1%)		
4	117 (18.8%)	156 (14.8%)		88 (40.0%)	42 (19.1%)				8 (11.1%)		
5	91 (14.6%)	221 (21.0%)		37 (16.8%)	56 (25.5%)				44 (61.1%)		
5. Did he social distancing during the pandemic affect your relationships with your colleagues?											
1	117 (18.8%)	364 (39.4%)	<0.001	33 (15.0%)	93 (42.3%)	<0.001	<0.001	<0.001	20 (27.8%)	0.032	0.149
2	52 (8.3%)	52 (5.6%)		30 (13.6%)	21 (9.5%)				9 (12.5%)		
3	221 (35.4%)	234 (25.4%)		37 (16.8%)	10 (4.5%)				13 (18.1%)		
4	117 (18.8%)	130 (14.1%)		54 (24.5%)	64 (29.1%)				13 (18.1%)		
5	117 (18.8%)	143 (15.5%)		66 (30.0%)	32 (14.5%)				17 (23.6%)		
6. Did social distancing during the pandemic affect your relationships with your friends?											
1	130 (20.8%)	364 (39.4%)	<0.001	55 (25.0%)	105 (47.5%)	<0.001	<0.001	0.001	20 (27.8%)	0.156	<0.001
2	117 (18.8%)	130 (14.1%)		12 (5.5%)	31 (14.0%)				17 (23.6%)		
3	156 (25.0%)	143 (15.5%)		34 (15.5%)	11 (5.0%)				10 (13.9%)		
4	130 (20.8%)	156 (16.9%)		85 (38.6%)	41 (18.6%)				12 (16.7%)		
5	91 (14.6%)	130 (14.1%)		34 (15.5%)	33 (14.9%)				13 (18.1%)		
7. Did social distancing during the pandemic affect your relationship with your family?											
1	429 (68.8%)	546 (59.2%)	<0.001	88 (40.0%)	163 (74.1%)	<0.001	<0.001	<0.001	36 (50.0%)	<0.001	<0.001
2	91 (14.6%)	130 (14.1%)		31 (14.1%)	41 (18.6%)				20 (27.8%)		
3	39 (6.3%)	91 (9.9%)		35 (15.9%)	5 (2.3%)				12 (16.7%)		
4	52 (8.3%)	104 (11.3%)		33 (15.0%)	0 (0.0%)				0 (0.0%)		
5	13 (2.1%)	52 (5.6%)		33 (15.0%)	11 (5.0%)				4 (5.6%)		
8. Was it more difficult to return to on-site learning and teaching for you?											
1	195 (31.3%)	182 (19.7%)	<0.001	73 (33.2%)	64 (29.1%)	<0.01	<0.001	<0.001	32 (44.4%)	0.003	0.079
2	91 (14.6%)	104 (11.3%)		0 (0.0%)	30 (13.6%)				0 (0.0%)		
3	78 (12.5%)	221 (23.9%)		43 (19.5%)	20 (9.1%)				6 (8.3%)		
4	91 (14.6%)	273 (29.6%)		52 (23.6%)	63 (28.6%)				14 (19.4%)		
5	169 (27.1%)	143 (15.5%)		52 (23.6%)	43 (19.5%)				20 (27.8%)		
9. Is it more difficult for you to interact now, after the online experience, with patients/teachers/colleagues?											
1	325 (52.1%)	286 (31.0%)	<0.001	73 (33.2%)	117 (53.2%)	<0.001	<0.001	<0.001	36 (56.3%)	0.097	<0.001
2	130 (20.8%)	156 (16.9%)		20 (9.1%)	23 (10.5%)				17 (26.6%)		
3	52 (8.3%)	117 (12.7%)		38 (17.3%)	22 (10.0%)				6 (9.4%)		
4	52 (8.3%)	234 (25.4%)		42 (19.1%)	16 (7.3%)				5 (7.8%)		
5	65 (10.4%)	130 (14.1%)		47 (21.4%)	42 (19.1%)				0 (0.0%)		

*p*-value < 0.05 was considered to be significant. !—tests not consistent.

## Data Availability

Not applicable.

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
