# Peer review of "The Impact of the COVID-19 Pandemic on Quality Education of the Medical Young Generation"

_ijerph, 2023, doi:10.3390/ijerph20053953_

Round 1

Reviewer 1 Report

The title of the manuscript is clear and of an adequate length.

The abstract has an adequate and functional structure. It provides basic elements on the purpose of the study, the method, the results and the discussion. However, it would be desirable for them to provide specific elements about the sample and the psychometric properties of the instrument used for data collection.

The keywords are functional and consistent with the purpose of the study.

The theoretical-conceptual introduction and contextualization is superficial. It requires a significant improvement to contextualize the way they are understanding “quality education” and what it means with the younger generations of doctors. In the same way, it would be pertinent to contextualize the place or country from which the research is built.

The method is constructed in a suitable way. The indication to provide more specific elements about the psychometric properties of the instrument used for data collection is reiterated. In the same way, it would be desirable that they provide elements related to the ethical criteria and scientific rigor that were met in the study. On the other hand, what is related to data analysis is somewhat superficial.

The presentation of results abuses the use of tables. They do not provide a summary explanation of their findings or of the elements that might stand out from the tables presented. The section requires significant review and improvement.

The discussion is built on the basis of their findings, but lacks theoretical and conceptual depth. A critical and reflective look at their findings is not appreciated. The section requires an improvement in terms of its writing and bibliographical updating. The emotional sphere in online classes and the relational dynamics that are built there are treated superficially.

The conclusions are flawed. It requires a revision and structural improvement, trying to give an explicit response to the original purpose of the study. On the other hand, it is recommended that the authors provide specific elements about the limitations and projections of the study.

A thorough and systematic review of the references section is recommended, trying to comply with the editorial standards of the journal.

Reviewer 2 Report

The paper was well designed and provided adequate information for publication.  Suggestions for paper revision are:

1. The writing style in the Introduction section is rather a list of related statements. Please try to make them in to more meaningful paragraphs, such as you can merge into the background and impacts of COVID-19, the teaching and learning paradigm under the pandemic situation, the introduction about the university's medical studies, and the lack of previous research on the topic.

On line 77, Because there are few studies.... Could the authors identify those studies in this section so that the lack of previous research on the topic can be clarified.

2. Lines 197-202, the authors referred to Planas et al. study which discussion on the issues about background of students such as incomes, access to technology, and learning environment.  I think this paper did not have any statistical analysis on the comparison of students' background concerning the mentioned data, only the preclinical and clinical status were compared.  The authors may make it clear and relevant in this paragraph.
